# PSMDet: Enhancing Detection Accuracy in Remote Sensing Images Through Self-Modulation and Gaussian-Based Regression

**DOI:** 10.3390/s25051285

**Published:** 2025-02-20

**Authors:** Jiangang Zhu, Yang Ruan, Donglin Jing, Qiang Fu, Ting Ma

**Affiliations:** 1School of Computer Science, Civil Aviation Flight University of China, Guanghan 618307, China; zhujiangang@cafuc.edu.cn (J.Z.); csfuqiang@cafuc.edu.cn (Q.F.); mating@cafuc.edu.cn (T.M.); 2Shanghai Aerospace Control Technology Institute, Shanghai 201109, China; ruanyang1987@163.com; 3Research and Development Center of Infrared Detection Technology, China Aerospace Science and Technology Corporation, Shanghai 201109, China

**Keywords:** object detection, remote sensing, multi-scale feature extraction, Gaussian regression loss, deep learning

## Abstract

Conventional object detection methods face challenges in addressing the complexity of targets in optical remote sensing images (ORSIs), including multi-scale objects, high aspect ratios, and arbitrary orientations. This study proposes a novel detection framework called Progressive Self-Modulating Detector (PSMDet), which incorporates self-modulation mechanisms at the backbone, feature pyramid network (FPN), and detection head stages to address these issues. The backbone network utilizes a reparameterized large kernel network (RLK-Net) to enhance multi-scale feature extraction. At the same time, the adaptive perception network (APN) achieves accurate feature alignment through a self-attention mechanism. Additionally, a Gaussian-based bounding box representation and smooth relative entropy (${\mathrm{smooth}}_{\mathrm{RE}}$) regression loss are introduced to address traditional bounding box regression challenges, such as discontinuities and inconsistencies. Experimental validation on the HRSC2016 and UCAS-AOD datasets demonstrates the framework’s robust performance, achieving the mean Average Precision (mAP) scores of 90.69% and 89.86%, respectively. Although validated on ORSIs, the proposed framework is adaptable for broader applications, such as autonomous driving in intelligent transportation systems and defect detection in industrial vision, where high-precision object detection is essential. These contributions provide theoretical and technical support for advancing intelligent image sensor-based applications across multiple domains.

## 1. Introduction

Object detection (OD) in ORSIs is fundamental to the automated interpretation of remote sensing (RS) data, facilitating the quick and precise identification and localization of significant items within extensive visible light images. This technology has exhibited numerous potential uses in agricultural monitoring, resource exploitation, marine law enforcement, and aviation security. With the rising appeal of high-resolution ORSIs and the swift advancement of computer capacity, OD approaches utilizing convolutional neural networks (CNNs) [1] have progressively emerged as the predominant study focus in this domain.

Rotated object detection (ROD) has evolved from conventional horizontal object detection (OD) methods. Existing horizontal OD techniques can be mainly categorized into two main groups: two-stage detectors and one-stage detectors. Two-stage detectors (e.g., R-CNN series [2,3,4,5,6]) rely on the region candidate network (RPN) [6] to generate a high-quality candidate region box before predicting the object location, which performs particularly well in scenarios with high accuracy requirements. In contrast, one-stage detectors (e.g., SSD [7], YOLO [8], and RetinaNet [9]) predict the object location directly from preset anchor boxes, which better meets real-time requirements.

Due to its unique vertical perspective, ORSIs usually contain rich spatial features and texture information. As shown in Figure 1, the objects in the images present diverse morphologies, which are characterized by arbitrary orientation, multi-scale, and high aspect ratio. This complexity poses a great challenge to the ROD task. Early research on rotational detection [10,11] attempted to enrich the diversity of training samples by introducing rotational transformations to enhance the robustness of the model in the OD task. Ma et al. [12] further proposed to enhance the model performance by increasing the rotation angle parameter and introducing redundant anchor boxes. However, although these approaches have made some progress in detection accuracy, they simultaneously significantly increase the complexity and computational burden of the model, thus limiting its feasibility in practical application scenarios.

To solve the above problems, researchers have gradually explored new technical paths. For example, some approaches try to transform traditional horizontal anchor boxes into oriented anchor boxes and combine them with region alignment techniques to model rotational invariance. Such methods [13,14,15,16] are gradually becoming mainstream because they are able to capture the rotational features of the object more efficiently, thus improving the detection accuracy. However, despite the breakthroughs made by these techniques in the field of ROD, they are still unable to escape the limitations of the traditional detection paradigm completely.

In the field of RSOD, the main problems faced by the traditional detection paradigm can be summarized into three levels of misalignment:Spatial Misalignment [17]: Researchers usually deepen the network hierarchy by stacking a large number of 3×3 small convolutional kernels to extend the theoretical Receptive Field (RF). However, studies have shown that it is difficult to significantly improve the Effective Receptive Field (ERF) [18] even when stacking multiple layers of small convolutional kernels. This limitation restricts the model from effectively utilizing the contextual information around the object, thus affecting the multi-scale feature extraction capability. In particular, the model exhibits large deficiencies in terms of scale adaptability and resistance to background interference.Feature Misalignment: FPN aims to reduce the semantic differences between different feature layers by fusing multi-scale features. However, the traditional model is more limited in scale-perception ability when the scale of the objects in the detection scene varies greatly. Meanwhile, the spatial perception ability is also deficient for objects with complex shapes, rotations, and position changes. To adapt to different object representations (e.g., bounding boxes [19], centroids [20], vertices [21], etc.) and their unique optimization objectives and constraints, the model also needs to have higher task-perception capability. However, due to the lack of perception in the above three aspects, the model is challenging in providing the necessary scale, space, and task-sensitive features for the detection head, which restricts the further improvement of the detection performance.Regression Misalignment: Regarding object representation, researchers commonly use a method based on five parameters (x,y,w,h,θ) to represent a rectangular oriented bounding box (OBBox). However, this representation leads to problems such as discontinuities in the regression loss and square-like objects. Meanwhile, the following of the traditional regression loss function $L_{n-\mathrm{norm}}$ (commonly used in ROD ${\mathrm{smooth}}_{L_{1}}$ loss [3]) leads to a lack of effective coupling mechanism of the five parameters during training, resulting in an inconsistency between the regression loss and the metric—even a low loss does not result in a high IoU. If these issues are not effectively addressed, the detection result will show the problem of angle prediction deviation of the object [22] (as shown in Figure 2 (left)).

To tackle the challenges above, this work introduces a novel detection framework called Progressive Self-Modulation Detector (PSMDet), which incorporates self-modulation mechanisms at three critical stages: the backbone network, the FPN, and the detection head, thereby effectively addressing the limitations inherent in conventional detection paradigms for RSOD. The distinct contributions of this work might be encapsulated as follows:We propose a novel backbone network, which is the Reparameterized Large Kernel Network (RLK-Net). Based on a large convolutional kernel and combined with a structural reparameterization technique, the network aims to extend the ERF, hence enhancing its capacity to acquire contextual information and markedly improving the extraction of multi-scale features.We propose an adaptive perception network (APN) that follows the FPN and self-regulates the fused features of the upper and lower layers of the FPN through a self-attention mechanism. Specifically, the network can achieve adaptive perception at the feature level, spatial location, and output channel, achieving precise feature alignment at the spatial, scale, and task levels.We introduce a GD-based BBox representation and devise a new ${\mathrm{smooth}}_{\mathrm{RE}}$ regression loss function to address the problems of loss discontinuity and the square-like object caused by the five-parameter BBox representation as well as the regression inconsistency caused by the $L_{n-\mathrm{norm}}$ regression loss [22] (as shown in Figure 2 (right)).

Through experimental validation, the PSMDet method proposed in this study achieves 90.69% and 89.86% mAP on two public datasets, HRSC2016 and UCAS-AOD, respectively, which demonstrates its excellent and robust performance. These results validate the proposed method and offer theoretical insights and technical support for further RSOD research. Moreover, the proposed method demonstrates intense multi-scale feature extraction and alignment capabilities. It also holds potential for broader applications, such as autonomous driving in intelligent transportation systems and defect detection in industrial vision tasks.

## 2. Related Work

In the field of ROD, achieving the ability to recognize objects with arbitrary angles has been an important research topic of great interest. Researchers have mainly tackled this challenge by adapting existing network models. To this end, a common approach [10,11] is to utilize rotational transformation techniques to augment and expand the training samples, thereby improving the model’s robustness in the ROD task. However, this method still relies on the horizontal BBox representation, thus making it difficult to accurately fit the shape of the rotated object, which ultimately affects the detection results.

With the continuous progress of technology, the two-stage OD model gradually shows its unique advantages. In this framework, by introducing the rotation angle θ as a regression parameter, it is able to locate the rotated rectangular box more effectively.

Currently, most of the research work focuses on detection methods based on OBBox. The following section summarizes and analyzes the development routes, the core problems faced, and the potential countermeasures for solving them in this field.

### 2.1. Development of OBBox-Based OD Algorithms

In the field of arbitrary orientation text detection, Ma et al. [12] proposed the first rotated candidate box detection algorithm based on RPN architecture, RRPN (Radar Region Proposal Network). The model generates candidate regions by presetting anchor boxes with different scales and rotation angles, extracts features using RROI pooling and classifies them through a fully connected layer. However, due to the limited preset angles, it is difficult for RRPN to cover all possible object directions fully. To solve this problem, Jiang et al. [23] proposed the $R^{2}$-CNN, which focuses on the prediction of positional parameters by improving the BBox representation and adding the second-stage prediction branch, and, at the same time, optimizes the RoI pooling layer so as to capture the image features better. The $R^{2}$-CNN further expands the detection idea based on RRPN, which makes the detection more flexible and better. $R^{2}$-CNN further expands the detection idea based on RRPN, which makes the detection more flexible and accurate.

On this basis, the RoI Transformer algorithm proposed by Ding et al. [13] realizes the accurate prediction of rotated candidate boxes by predicting the regression parameters of rotated anchor boxes through horizontal anchor boxes. The method does not need to increase the number of anchor points, which significantly improves the detection efficiency and, at the same time, effectively combines the advantages of traditional horizontal detection and rotated detection. Yang et al. [24,25] proposed an AO-RCNN model on this basis, which further migrates the ideas of $R^{2}$-CNN and RRPN and introduces the Adaptive RNN model. They migrated and introduced the Adaptive ROI Align mechanism, which reduces the redundant noise in the detection process. In addition, APE [26] innovatively uses periodic 2D vectors to represent the angles, which simplifies the representation of the OBBox and improves the stability of the model.

$R^{3}$Det [14] achieves end-to-end single-stage detection, which breaks through the bottleneck of the traditional two-stage model and provides a faster and more accurate ROD scheme. ICN [27] further enhances the ability to capture objects with different angles during detection by combining image cascade, FPN, and multi-scale convolutional kernels. $S^{2}$A-NET [15] mitigates the inconsistency between regression and classification by combining the feature-aligned convolution with the ROD module. Finally, the ReDet algorithm proposed by Han et al. [16] realizes efficient feature extraction for rotated objects based on rotation-equivariant networks and introduces the rotation-invariant RoI alignment technique to improve the stability and accuracy of detection further.

Despite the significant progress made by the above methods in the field of ROD, they still face the limitation of parameter representation and regression loss function, which leads to the problem of misalignment of the OBBox regression, which in turn affects the training stability of the model and its detection performance.

### 2.2. Analysis of the Limitations of OBBox

The OBBox usually adopts a five-parameter representation (x,y,w,h,θ). According to the different value ranges of θ, it can be categorized into two representations: the OpenCV definition $D_{oc}$ and the long edge definition $D_{le}$, and one of the two representations is required when designing the model. Despite its simplicity, the five-parameter representation has three major limitations in practical applications:Loss Discontinuity [28]. Due to the periodicity in the definition of the angle parameter, the failure of the angle to regress toward the optimal clockwise direction when the anchor or proposal is located in either the horizontal or vertical direction (θ defined boundaries) may lead to dramatic fluctuations in the loss function. This phenomenon mainly stems from the mismatch between the five-parameter representation and the traditional loss function ${\mathrm{smooth}}_{L_{1}}$, which in turn reduces the stability of the model in dealing with these special cases. This problem exists in both the $D_{oc}$ and $D_{le}$ representations.Square-like Objects [22]. For objects that are close to square, $D_{le}$-based networks often exhibit high IoU but high loss when the metric method is inconsistent with the loss calculation. This situation causes the model to encounter difficulties in parameter regression, thus affecting the overall detection performance. This problem reflects the inherent limitations of the five-parameter representation in the detection accuracy of square-like objects.Regression Inconsistency [22]. Angle, width, height, and center coordinates in the five-parameter representation use different measurement units, respectively, and thus they exhibit different functional relationships with IoU (as shown in Figure 3 diagram). This difference leads to inconsistency in the regression process, further reducing the convergence of training and detection accuracy. It may be difficult for the model to optimize these parameters simultaneously at different training stages, leading to undesirable detection results even under IoU conditions.

### 2.3. Loss Function-Based Optimization Strategy

According to the above analysis, the limitation of the traditional loss function ${\mathrm{smooth}}_{L_{1}}$ may lead to regression difficulties. Therefore, the improvement of the loss function has become an important direction in ROD research. Yang et al. [31] proposed the SCRDet model, which combines the IoU with the ${\mathrm{smooth}}_{L_{1}}$ loss, and designs the IoU-${\mathrm{smooth}}_{L_{1}}$ loss to mitigate loss fluctuations during angular regression. Subsequently, SCRDet++ [32] further improved the ${\mathrm{smooth}}_{L_{1}}$ loss, and Chen et al. [33] proposed the PIoU loss, which improves the accuracy of the rotational angular regression and the IoU simultaneously. In addition, constraint loss [34] introduces a deblocking modulation mechanism to solve the problem of abrupt loss changes effectively.

The researchers also used GD to represent the OBBox implicitly and designed the associated loss function based on it. Yang et al. [22] proposed the SkewIoU loss for approximate non-differentiable learning by using GWD (Gaussian Wasserstein Distance), which effectively solves the problems of loss discontinuity, square-like object, and regression inconsistency. In order to overcome the shortcomings of GWD, which requires additional hyperparameter adjustment, Yang et al. [35] further proposed a Kalman filter-based KFIoU loss function, which further improves the training stability and detection performance of the model.

## 3. Methodology

As shown in Figure 4, we propose a novel OD method, PSMDet, which is designed to meet the special scenario requirements of OD in ORSIs.The architecture of PSMDet is mainly composed of three core components: Backbone, Neck, and Head. The Backbone is responsible for multi-scale feature extraction; the Neck part combines FPN and ASN to realize multi-level feature fusion and alignment; and the Head part is configured with five detector heads to cope with the classification and regression tasks of objects at different scales.

In the Backbone section, we adopt RLK-Net as the backbone for feature extraction. The network extracts contextual information with larger ERF through a parallel multi-kernel strategy to obtain more robust scale-sensitive features without the need for an overly deep network structure. This design generates hierarchical feature maps with different semantic levels through layer-by-layer downsampling. Specifically, shallow features capture coarse-grained spatial information such as texture and edges, which are crucial for small objects. In contrast, intermediate and deep features provide more detailed semantic information, especially those related to object categories, making these layers more suitable for the task of detecting medium-to-large-sized objects.

In the Neck section, we combine the FPN with the proposed ASN. Firstly, the spatial resolution of the Backbone features from different layers is recovered by the upsampling operation, which is followed by splicing in the channel dimension to realize the effective fusion of the different scale features with the contextual information. The ASN, on the other hand, enhances the scale-perception, spatial-perception, and task-perceptive capabilities of the model by introducing multiple self-attention mechanisms among feature layers, spatial locations, and output channels, thus further improving the discriminative representation of Neck features. This multi-level feature integration significantly improves the model’s adaptability to complex scenes.

In the Head section, PSMDet is configured with five detection heads, each of which is responsible for the classification and anchor regression tasks of different scale objects through two decoupled branches. This structural design enables the model to cope with the demand of multi-scale OD flexibly. In addition, we introduce an OBBox representation method using GD modeling and design the ${\mathrm{smooth}}_{\mathrm{RE}}$ regression loss function, which effectively solves the problems of loss discontinuity, square-like objects, and regression inconsistency, and thus significantly improves the overall detection performance.

### 3.1. RLK-Net Backbone Architecture

ORSIs are usually taken at high resolution from a vertical perspective, which leads to significant differences in the scale of feature objects. In order to better interpret the shapes and orientations of these objects, it is crucial to acquire contextual information about the background and neighboring environments that are closely related to them at the same time. The acquisition of such information is important for improving the accuracy of OD. However, traditional CNNs usually consist of a stack of convolutional kernels with a fixed RF. When the network structure is shallow, it is difficult to achieve a global understanding of the object scene, and detection methods that rely only on local features often fail to achieve the desired accuracy when dealing with large-scale objects. Although the RF can be theoretically extended by increasing the stacking of convolutional kernels, this approach not only significantly increases the computational complexity, but also the actual ERF may not be improved accordingly [18]. We recognize that atrous convolution (AC) and attention mechanisms have been widely used to enhance global perception. However, while AC kernels need to have a larger receptive field, an increase in the dilatation rate also leads to a gradual sparsification of the feature space. Meanwhile, while the attention mechanism with global perception capability is effective, its square-scale computational complexity may be challenging to deploy on lightweight devices. To solve this problem, we propose the RLK-Net backbone network, employing a significant kernel mechanism similar to RepLKNet [36] and SLaK [37] to achieve global perception, which has been shown to achieve comparable or even better results than the attention mechanism in classification or other downstream tasks while avoiding the high computational complexity problem associated with the attention mechanism.

As shown in Figure 5, the overall structure of the RLK-Net backbone network is divided into four stages connected by downsampling blocks. Specifically, the first downsampling block uses two 3×3 convolutional layers with a stride of 2 to convert the raw input into a feature map with a channel number of *C*, where *C* is a hyperparameter of the architecture. The remaining three downsampling blocks are each channel-extended using a 3×3 convolutional layer with a stride of 2, resulting in four stages with channel counts of $\left\{C,2C,4C,8C\right\}$, respectively. Our channel assignment strategy follows the design of a 2× increment at each stage of the ResNet [38] architecture. Since the input image size used in our experiments is 800×800, this conflicts with the “composite scaling” channel assignment strategy of the EfficientNet [39] architecture based on the input resolution. Therefore, the architecture design of the RLK-Net does not consider the “composite scaling” scheme.

Each stage of RLK-Net consists of a fundamental building block similar to the ConvNeXt V2 block [40], i.e., the DW convolutional layer and the Convolution Feedforward Network (ConvFFN) with Global Response Normalization (GRN) unit. The overall design of RLK-Net replaces the ConvNeXt V2 Block with our fundamental building block—the Large Kernel Block (LK) and Small Kernel Block (SK) blocks. In the LK block, we introduce the Sparse Feature Enhancement (SFE) block, the Squeeze-and-Excitation (SE) block [41], the ConvFFN layer, and the BN layer. Using SE blocks, we can achieve efficient communication and spatial aggregation between feature channels, thus enhancing the network’s recognition capability at different scales. The structural design optimizes the network’s overall performance, enabling it to accurately image cascade and understand feature objects at different scales when processing high-resolution aerial images. We chose to use BatchNorm (BN) [42] instead of LayerNorm (LN) [43] to improve the efficiency of the model because the BN can be equivalently merged into a convolutional layer, thus eliminating the inference cost. In addition, the second BN layer added after the ConvFFN is also integrated into the previous layers. For the SK block, we use the 3×3 Depthwise (DW) convolution to replace the SFE block in the LK block. By combining the design of the LK block and SK block, we realize the efficient extraction and expression of features at different scales.

As shown in Table 1, the architectural hyperparameters of the two RLK-Net variants (T/S) used in this study include the block number $\left\{D_{1},D_{2},D_{3},D_{4}\right\}$ of fundamental building blocks and the number $\left\{C_{1},C_{2},C_{3},C_{4}\right\}$ of channels for the four stages. The stacking of RLK-Net follows the original design of the ConvNeXt V2-T/S, where *C* = 96 and *N* = (3, 3, 9, 3). By default, Stage 1 uses an SK block of 3×3, while the last three stages use 13×13 LK blocks to expand the RF and improve the feature extraction performance. In designing Stage 3 of RLK-Net, our strategy of alternately stacking LK blocks and SK blocks draws on the idea of UniRepLKNet [44] (as shown in Figure 5 and Table 1), such as “9 + 9” means that Stage 3 contains 9 LK blocks and 9 SK blocks. “9 + 18” means there are 9 LK blocks and 18 SK blocks. This strategy is because SK blocks are good at extracting fine-grained features such as edges and textures; they are more effective in capturing local patterns, further enriching and enhancing the global features extracted by LK blocks, and improving the abstraction level of spatial patterns. In addition, more learnable parameters and nonlinearities are introduced through the SK blocks to deepen the model appropriately, improving the model’s representational ability but not leading to excessive parameters.

### 3.2. Sparse Feature Enhancement Block

From the structural reparameterization [45,46,47,48,49,50] perspective, the AC layer can be viewed as a non-AC layer with a large sparse kernel. Thus, the entire reparameterized block can be effectively viewed as a single large-kernel convolution. For example, taking *K* = 13 as an example, for larger values of *K*, more AC layers can be introduced further to enhance the model’s expressiveness and feature extraction capabilities.

In the current research, combining large kernel convolution with parallel small kernel convolution is widely recognized because small kernel convolution can effectively capture fine-scale patterns to enhance the feature representation during training [36]. Precisely, the outputs of these two types of convolution are summed after their respective backward BN layers. After the model training is completed, we use structural reparameterization to integrate the BN layer into the convolutional layer so that the small kernel convolution can be merged with the large kernel convolution equivalently in the inference stage. This process optimizes the model’s computational efficiency and improves the quality of feature extraction. Enhancing the ability of the large kernel convolution to capture sparse patterns is particularly critical in our design. A sparse pattern is defined as a pixel on the feature map that may be more closely related to its distant pixels, not only dependent on neighboring pixels. The need to capture such complex spatial relationships is highly compatible with the mechanism of AC.

Under the sliding window perspective, a AC layer with a dilatation rate of *r* can effectively capture the spatial patterns between neighbouring pixels that are at a distance of r−1 from each relevant pixel. Therefore, we design a parallel AC layer to the large kernel convolution and sum up the outputs of both. This parallel convolution structure aims to enhance the model’s ability to understand complex image features and is particularly suitable for RS images with significant spatial dependence and long-range correlation.

In order to effectively alleviate the inference burden associated with the introduction of an AC layer, we propose an innovative strategy that equivalently converts a complex SFE Block structure into a single non-AC layer for efficient inference. This conversion is based on a central observation: ignoring certain pixels in the input data (i.e., not weighting and summing these pixels) in an AC is equivalent to inserting additional zero-padding in its convolutional kernel. This approach allows mapping a smaller-sized but AC kernel into an equivalent larger-sized and sparse non-AC kernel. Let the original kernel size of the AC layer be k, and the kernel size of the equivalent non-AC layer can be computationally determined to be (k−1)r+1, which is a value referred to as the equivalent kernel size.

Moreover, we find that the transformation from the original AC kernel $F\in R^{k\times k}$ to the equivalent non-AC kernel $F^{\prime }\in R^{\left((k-1)r+1\right)\times \left((k-1)r+1\right)}$ can be implemented succinctly and efficiently by combining a transposed convolution operation with a stride of *r*, and a special identity kernel $I\in R^{1\times 1}$ is combined concisely and efficiently. In the implementation, it can be represented in pseudo-code similar to PyTorch-like style:(1)$$F^{\prime }=\mathrm{conv}\_\text{transpose2d}(F,I,\mathrm{stride}=r).$$

For any given F and number of input channels, the results obtained from AC operation using F and a specific dilatation rate *r* are identical to those obtained from non-AC operation using the transformed $F^{\prime }$. Based on this finding, we propose the Sparse Feature Enhancement (SFE) block, which combines a non-atrous small-kernel convolutional layer with multiple atrous small-kernel layers to enhance the performance of the non-atrous large-kernel convolutional layer. The flexibility of the design is reflected in the choice of hyperparameters, including kernel size *K*, small kernel size *k*, and dilatation rate *r*, satisfying the condition (k−1)r+1≤K.

As shown in Figure 6, set *K* = 13 (the default setting in the experiments in this study). We use five parallel convolutional layers, where the kernel size k and the dilation rate r of each layer are (5,7,3,3,3) and (1,2,3,4,5), respectively, yielding an equivalent kernel size of (5, 13, 7, 9, 11). For inference, we convert the SFE block into a single non-AC layer. First, the BNs of each branch are merged into the previous convolutional layer to reduce computational redundancy; next, each AC layer with dilation rate *r* > 1 is converted into an equivalent non-AC layer using Equation (1). Finally, after an appropriate zero padding is performed to ensure that the output is consistent with the expected large kernel, the convolutional kernels of all the transformed non-AC layers are summed. For example, for the convolutional layer k=3,r=3 in Figure 6, it needs to be converted to a sparse kernel of 7×7 and then zero-padded with 3 pixels around it.

### 3.3. Adaptive Perception Network

Dimension design. We must adjust the feature dimensions to realize the adaptive perception of multi-scale fused features in scale, space, and task dimensions. Consider the spliced feature tensor generated by the $X_{in}={\left[X_{i}\right]}_{i=1}^{L}$, where *L* represents the total number of layers of the object. By upsampling or downsampling techniques, we ensure that the feature sizes of all levels match those of the intermediate levels to construct a four-dimensional feature tensor $X\in R^{L\times H\times W\times C}$, where *H*, *W*, and *C* correspond to the heights, widths, and number of channels of the features, respectively. For subsequent processing, we reshape X into a 3D tensor $X\in R^{L\times S\times C}$ where S=H×W. Based on this representation, we further explore the role of each tensor dimension.

Given a feature tensor $X\in R^{L\times S\times C}$, the generalized formulation of the self-attention mechanism can be expressed as(2)F(X)=Attn(X)·X.

This formulation demonstrates how features can be weighted through a self-attention mechanism that enhances important features and suppresses redundant information to improve the model’s performance in complex tasks.

Decomposition and realization of the self-attention mechanism. When applying the self-attention mechanism on the feature tensor $X\in R^{L\times S\times C}$, the traditional methods may face a computational bottleneck due to the high dimensionality. To solve this problem, we propose an innovative attention decomposition strategy that decomposes the attention function Attn(·) into three independent sub-attention functions for the scale (*L*), space (*S*) and task (*C*) dimensions, respectively:(3)$$F\left(X\right)={\;\mathrm{Attn}}_{C}\left({\mathrm{Attn}}_{S}\left({\mathrm{Attn}}_{L}\left(X\right)\cdot X\right)\cdot X\right)\cdot X,$$
where ${\mathrm{Attn}}_{L}\left(\cdot \right)$, ${\mathrm{Attn}}_{S}\left(\cdot \right)$ and ${\mathrm{Attn}}_{C}\left(\cdot \right)$ correspond to the attention function of the scale, space, and task dimensions (as shown in Figure 7), respectively. Through this decomposition, we are able to effectively reduce the computational complexity while fully capturing the correlation of the feature tensor in different dimensions.

Scale-perception attention ${\mathrm{Attn}}_{L}$. We design the scale-perception attention mechanism to fuse features of different scales dynamically. This mechanism adjusts the weight of each scale feature by calculating its semantic importance. Specifically, we use global average pooling (on S×C dimensions) to aggregate features, which are then linearly transformed by $F^{1\times 1}$ convolutional layers (as a linear function f(·)) and applied with a hard S-shaped function $\sigma \left(x\right)=\max\left(0,\min\left(1,\frac{x+1}{2}\right)\right)$ to normalize and finally obtain the scale attention weights, which are applied to the original features:(4)$${\mathrm{Attn}}_{L}\left(X\right)\cdot X=\sigma \left(f\left(\frac{1}{S\times C}\sum_{s=1}^{S}\sum_{c=1}^{C}X_{:,s,c}\right)\right)\cdot X.$$

Space-perception attention ${\mathrm{Attn}}_{S}$. In order to capture discriminative regions at spatial locations, we design a space-perception attention module. Considering the high dimensionality of the spatial dimension, we adopt a two-step strategy. Firstly, we utilize deformable convolution [51] to achieve sparse learning of attention, and then we perform feature aggregation at the same spatial location across different scales. For each location and scale, we adjust the feature weights based on the spatial offset $\Delta p_{k}$ and importance scalar $\Delta m_{k}$, which was learned from the intermediate-level input features of X:(5)$${\mathrm{Attn}}_{S}\left(X\right)\cdot X=\frac{1}{L}\sum_{l=1}^{L}\sum_{k=1}^{K}w_{l,k}\cdot F^{DC}\left(l;p_{k}+\Delta p_{k};c\right)\cdot \Delta m_{k},$$
where *K* is the number of sparsely sampled locations, $w_{l,k}$ is the weight coefficient, and $F^{DC}$ denotes the deformable convolution.

Task-perceprion attention ${\mathrm{Attn}}_{C}$. In order to support multi-task learning and optimize the feature representation for different tasks, we design the task-perception attention mechanism. This mechanism selectively activates or suppresses feature channels by comparing different feature responses under the activation function. Specifically, we use two parallel activation paths and select the optimal response by the maximum function:(6)$${\mathrm{Attn}}_{C}\left(X\right)\cdot X=\max\left(\alpha^{1}\left(X\right)\cdot X_{c}+\beta^{1}\left(X\right),\alpha^{2}\left(X\right)\cdot X_{c}+\beta^{2}\left(X\right)\right),$$
where $X_{c}$ represents the *c*-th channel slice in the input feature matrix X and the parameter set ${\left[\alpha^{1},\alpha^{2},\beta^{1},\beta^{2}\right]}^{⊤}=\theta \left(\cdot \right)$ is a hyperfunction for learning and controlling activation thresholds. The design of this function is inspired by Dynamic ReLU [52], and it has been adaptively improved. Its implementation steps include global average pooling in L×S dimensions to reduce the feature dimensions and extract the global feature information; nonlinear mapping of the pooled features by two fully connected layers and the introduction of a normalization layer to improve the stability and convergence speed of the model; and, finally, the application of a shifted sigmoid function to restrict the output to the interval [−1,1], providing more flexible feature activation regulation for the attention mechanism.

The design of this function is inspired by Dynamic ReLU [52] and has been adaptively improved.

### 3.4. Relative Entropy Regression Loss

Traditional $L_{n-\mathrm{norm}}$ regression loss. Regression loss is an important component of most current OD algorithms. For horizontal BBox regression, the model outputs four main items such as position and size:(7)$$\begin{array}{cc} \left(\delta_{x}^{p},\delta_{y}^{p}\right) & \;=\left(\frac{1}{w_{a}}\left(x_{p},y_{p}\right)-\frac{1}{h_{a}}\left(x_{a},y_{a}\right)\right),\;\\ \left(\delta_{w}^{p},\delta_{h}^{p}\right) & \;=\log\left(w_{p},h_{p}\right)-\log\left(w_{a},h_{a}\right). \end{array}$$

To match the four parameters in the GT:(8)$$\begin{array}{cc} \left(\delta_{x}^{t},\delta_{y}^{t}\right) & =\left(\frac{1}{w_{a}}\left(x_{t},y_{t}\right)-\frac{1}{h_{a}}\left(x_{a},y_{a}\right)\right),\\ \left(\delta_{w}^{t},\delta_{h}^{t}\right) & =\log\left(w_{t},h_{t}\right)-\log\left(w_{a},h_{a}\right). \end{array}$$
where *x*, *y*, *w*, and *h* denote the center coordinates, width, and height, respectively. The subscripts *t*, *a*, and *p* denote the GT BBox, the anchor BBox, and the prediction BBox, respectively.

An extension to the ROD model requires additional processing of the angular parameter θ:(9)$$\delta_{\theta }^{p}=\frac{1}{\pi }\left(\theta^{p}-\theta^{a}+k\pi \right),\delta_{\theta }^{t}=\frac{1}{\pi }\left(\theta^{t}-\theta^{a}+k\pi \right),$$
where *k* is an integer ensuring that $\left(\theta^{t}-\theta +k\pi \right)\in \left[-\frac{\pi }{4},\frac{3\pi }{4}\right)$.

The overall regression loss for ROD is(10)$$L_{reg}=L_{n-\mathrm{norm}}\left(\Delta \delta_{x},\Delta \delta_{y},\Delta \delta_{w},\Delta \delta_{h},\Delta \delta_{\theta }\right),$$
where $\Delta \delta_{x}=\delta_{x}^{p}-\delta_{x}^{t}=\frac{x^{p}-x^{t}}{w_{a}},\Delta \delta_{y}=\delta_{y}^{p}-\delta_{y}^{t}=\frac{y^{p}-y^{t}}{h_{a}},\Delta \delta_{w}=\delta_{w}^{p}-\delta_{w}^{t}=\log\left(\frac{w_{p}}{w_{t}}\right),\Delta \delta_{h}=\delta_{h}^{p}-\delta_{h}^{t},\Delta \delta_{h}^{t}=\log\left(\frac{h_{p}}{h_{t}}\right)$ and $\Delta \delta_{\theta }=\delta_{\theta }^{p}-\delta_{\theta }^{t}=\frac{1}{\pi }\left(\theta^{p}-\theta^{t}\right)$.

This means that the OBB parameters are optimized independently, which makes the loss very sensitive to the lack of fit of any one parameter, resulting in detection results that are prone to angle deviation, as shown in Figure 2a. These are very detrimental to high-precision detection.

In traditional ROD methods, the OBBox adapts to the rotational characteristics of the object by adding angular parameters. However, as discussed in Section 2.2, the five-parameter representation of the OBBox presents significant challenges when dealing with loss discontinuities, square-like objects, and regression inconsistencies. To address these issues, the researcher proposed a new method to transform the OBBox into a 2D GBBox. As shown in Figure 8, this method realizes the OBBox by mapping the geometric features B(x,y,w,h,θ) to the parameters of a GBBox N(μ,Σ) for a flexible representation and optimization.

Specifically, the conversion process can be expressed as follows:(11)$$\begin{array}{cc} \mu & ={\left[x,y\right]}^{⊤},\\ \Sigma^{1/2} & =R\Lambda R^{⊤}=\left[\begin{array}{cc} \cos\theta & -\sin\theta \\ \sin\theta & \cos\theta \end{array}\right]\left[\begin{array}{cc} \frac{w}{2} & 0\\ 0 & \frac{h}{2} \end{array}\right]{\left[\begin{array}{cc} \cos\theta & -\sin\theta \\ \sin\theta & \cos\theta \end{array}\right]}^{⊤}, \end{array}$$
where R represents the rotation matrix, and Λ is the diagonal matrix of eigenvalues. Through this transformation, the Gaussian BBox is not only able to adapt to objects with arbitrary directions and shapes but also effectively mitigates the regression misalignment problem caused by the OBBox parameterization in the traditional method.

Relative Entropy (RE): Inspired by the IoU-based loss [29,30], we propose a new regression loss based on RE. The RE between two 2D GDs is defined as(12)$$\mathrm{RE}\left(N_{g}∥N_{p}\right)=\underset{\mathrm{chain}\;\mathrm{coupling}\;\mathrm{of}\;\mathrm{all}\;\mathrm{parameters}\;}{\underset{︸}{\frac{1}{2}{\left(\mu_{p}-\mu_{g}\right)}^{⊤}\Sigma_{p}^{-1}\left(\mu_{p}-\mu_{g}\right)+\frac{1}{2}\mathrm{Tr}\left(\Sigma_{p}^{-1}\Sigma_{g}\right)+\frac{1}{2}\log\frac{\left|\Sigma_{p}\right|}{\left|\Sigma_{g}\right|}}}-1.$$

In the RE loss, the $\mathrm{RE}\left(N_{g}\left|\right|N_{p}\right)$ terms contain the shape parameter Σ and the center parameter μ, forming a chained coupling relationship. This optimization mechanism allows the parameters to be co-optimized, thus facilitating high-precision detection.

The specifics unfold as follows: (13)$${\left(\mu_{p}-\mu_{g}\right)}^{⊤}\Sigma_{p}^{-1}\left(\mu_{p}-\mu_{g}\right)=\frac{4{\left(\Delta x\cos\theta_{p}+\Delta y\sin\theta_{p}\right)}^{2}}{w_{p}^{2}}+\frac{4{\left(\Delta y\cos\theta_{p}-\Delta x\sin\theta_{p}\right)}^{2}}{h_{p}^{2}},$$(14)$$\mathrm{Tr}\left(\Sigma_{p}^{-1}\Sigma_{g}\right)=\frac{h_{g}^{2}}{w_{p}^{2}}\sin^{2}\Delta \theta +\frac{w_{g}^{2}}{h_{p}^{2}}\sin^{2}\Delta \theta +\frac{h_{g}^{2}}{h_{p}^{2}}\cos^{2}\Delta \theta +\frac{w_{g}^{2}}{w_{p}^{2}}\cos^{2}\Delta \theta ,$$(15)$$\ln\frac{\left|\Sigma_{p}\right|}{\left|\Sigma_{g}\right|}=\ln\frac{h_{p}^{2}}{h_{g}^{2}}+\ln\frac{w_{p}^{2}}{w_{g}^{2}},$$
where $\Delta x=x_{p}-x_{g},\Delta y=y_{p}-y_{g},\Delta \theta =\theta_{p}-\theta_{g}$. This formulation highlights the joint contribution of the center and shape parameters in accurately capturing the relationship between the predicted and GT box. By leveraging this coupled optimization, the RE loss addresses challenges in traditional regression approaches and offers a robust framework for high-precision OD.

RE Regression Loss. The training process for PSMDet is (1) predict the offset $\Delta \left(\delta_{x}^{p},\delta_{y}^{p},\delta_{w}^{p},\delta_{h}^{p},\delta_{\theta }^{p}\right)$; (2) decode the prediction offset into OBBox format; (3) encode OBBox into GBBox format; and (4) compute the RE loss.

As shown in Figure 3, when using RE directly as the loss function, it varies very drastically, causing the training to fail to converge. To smooth out the RE and enhance its expressiveness, we convert the RE into an IoU-like metric by means of a nonlinear transformation f(·):(16)$${\mathrm{smooth}}_{\mathrm{RE}}=1-\frac{1}{\tau +f(\mathrm{RE})},\;\tau \geq 1.$$

In this study, two transformations, $\sqrt{\mathrm{RE}}$ and log(RE), are explored. The hyperparameter τ is used to adjust the loss scale to adapt to different application scenarios. As shown in Figure 3, after this smoothing treatment, the curve trend of ${\mathrm{smooth}}_{\mathrm{RE}}$ is more consistent with the trend of the IoU loss curve and measures the distance of non-overlapping OBBoxs.

Multi-Task Loss Function: (17)$$L=\frac{\lambda_{1}}{N_{\mathrm{pos}}}\left(\sum_{i}L_{cls}\left(c_{i},l_{i}^{g}\right)\right)\;+\frac{\lambda_{2}}{N_{\mathrm{pos}}}\left(\sum_{i}1_{\left[l_{i}^{g}\geq 1\right]}L_{\mathrm{RE}}\left(p_{i},{\;g}_{i}\right)\right)\;,$$
where $\lambda_{1},\lambda_{2}$ is the balancing parameter, which takes the default value of [1,2], and $1_{\left[\cdot \right]}$ is the schematic function. $N_{\mathrm{pos}}$ denotes the positive samples, respectively. Quantity $c_{i}$ and $b_{i}$ are the predicted categories and locations, and $l_{i}^{g}$ and $g_{i}$ are the true categories and locations. The classification loss $L_{cls}$ uses focal loss.

## 4. Experimentation

### 4.1. Datasets

The HRSC2016 dataset [53], designed for ship detection, includes three primary categories, 27 subcategories, and 1061 images. The images are categorized into a training set containing 436 images, a validation set containing 181 images, and a test set with 444 images, totalling 2976 vessel instances. The dataset comprises images sized between 300×300 and 1500×900, predominantly exceeding 1000×600. The resolutions vary from 0.4 m to 2 m, rendering them appropriate for fine-grained OD applications. The imagegraphs were captured from six ports, encompassing vessels both at sea and along the coast, so guaranteeing the diversity and representativeness of the data.

The UCAS-AOD dataset [54], released in 2014 and augmented in 2015, is mostly utilized for the detection of aeroplanes and vehicles. The collection has 2420 images, comprising 600 aviation shots and 310 vehicle imagegraphs, along with counterexamples. The training, validation, and test sets comprise 755, 302, and 453 images, respectively, totalling 14,596 instances, which include 3210 aeroplanes and 2819 automobiles.

### 4.2. Evaluation Metrics

To scientifically and objectively assess the efficacy of OD methods for ORSIs, it is imperative to employ a range of quantitative evaluation metrics—Precision, Recall, mAP, Intersection over Union (IoU), and Frames Per Second (FPS)—which collectively appraise model performance across various dimensions, including accuracy, localization precision, and processing speed. In practical applications, appropriate assessment metrics can be chosen based on varying requirements and experimental goals with the evaluation criteria utilized in this work being mAP.

Precision is defined as the ratio of accurately predicted positive cases to the total expected positive cases, which was calculated as illustrated in Equation (18):(18)$$\;\mathrm{Precision}\;=\frac{\mathrm{TP}}{\mathrm{TP}+\mathrm{FP}},$$
where TP represents a true positive instance, while FP signifies a false positive instance.

Recall is derived from true samples and is defined as the ratio of correctly predicted positive cases to the total true positive cases, as computed in Equation (19):(19)$$\;\mathrm{Recall}\;=\frac{\mathrm{TP}}{\mathrm{TP}+\mathrm{FN}},$$
where FN signifies a false negative example.

mAP is a crucial metric for evaluating the effectiveness of OD algorithms, as it comprehensively reflects the detector’s accuracy and recall. Calculating mAP often entails multiple steps: initially, for each category, the area under its PR (Precision–Recall) curve, referred to as AP (Average Precision), is computed, as illustrated in Equation (20); subsequently, the AP values across all categories are averaged, culminating in the determination of mAP. mAP for the OD task involving N categories is calculated as demonstrated in Equation (21). (20)$$\mathrm{AP}=\int_{0}^{1}\;\mathrm{Precison}\left(t\right)dt,$$(21)$$\mathrm{mAP}=\frac{1}{N}\sum_{i=1}^{N}{\mathrm{AP}}_{i},$$

IoU is an index utilized to quantify the extent of overlap between the model’s predicted BBox and the GT box. A larger IoU value indicates a greater overlap between the predicted box and the GT box, resulting in enhanced localization accuracy of the model. The formula for computing the IoU is presented in Equation (22):(22)$$\mathrm{IoU}=\frac{\mathrm{Area}\;\mathrm{of}\;\mathrm{Intersection}}{\mathrm{Area}\;\mathrm{of}\;\mathrm{Union}}$$

FPS is a statistic for assessing the velocity of an OD algorithm, representing the number of image frames processed each second. A higher FPS number indicates a more rapid image processing by the model and enhances real-time performance. The calculation of FPS often relies on the processing duration of an individual image. The result is influenced by various aspects, including model complexity, computational resource performance (e.g., CPU, GPU), and the extent of image processing parallelization.

### 4.3. Implementation Details

Our experiments are based on the MMRotate [55] ROD framework (which integrates numerous SOTA ROD algorithms), and the R-RetinaNet detector was chosen as the baseline model.

This study employed the P3 to P7 layers of the FPN to detect multi-scale objects, establishing an anchor at each feature map point for regression purposes. The data augmentation approach solely comprises random flipping and random rotation. During the label assignment phase, the threshold for matching positive examples is established at 0.5 to guarantee the integrity of the detection results.

The ablation experiments employed the mAP measures established by PASCAL VOC 2007 to facilitate a fair comparison with alternative approaches and were conducted on the HRSC2016 dataset. This dataset includes vessels with changes in multi-scale objects and high aspect ratios, which are characteristics that render OD in ORSIs a significant difficulty.

The studies utilized a solitary RTX 2080Ti GPU, with a batch size of 2, employing the AdamW optimizer, an initial learning rate of $1\times 10^{-4}$, and a weight decay coefficient of 0.05. The training duration was 72, and the resolution of the input samples was uniformly set to 800×800 pixels.

### 4.4. Ablation Studies

In this study, the baseline architecture we selected covers the ResNet backbone network and the R-RetinaNet detector. Except for the RLK-Net-S/T comparison test, RLK-Net-S is used as the backbone network in all ablation studies.

Then, we conducted an ablation study of various combinations of PSMDet components on detection efficacy. Table 2 illustrates that the integration of the three elements of RLK-Net, ASN, and ${\mathrm{smooth}}_{\mathrm{RE}}$ markedly enhances the self-modulation capability throughout the detection process, hence augmenting overall detection performance. The experimental results indicate that the sole use of the baseline model, devoid of a self-modulation mechanism, hampers the high accuracy of RSOD. This yields a map of merely 85.37% for the baseline model.

The implementation of RLK-Net enhances detection performance by 2.17%, demonstrating that the RLK-Net backbone network effectively extracts more robust scale-sensitive features. Moreover, the ASN’s tri-layer adaptive perception system enhances detection performance by 1.22%. The implementation of ${\mathrm{smooth}}_{\mathrm{RE}}$ loss guarantees metric consistency with the loss, enhancing PSMDet’s detection performance by 1.93%, resulting in an overall improvement of 5.32% relative to the baseline model, thereby thoroughly substantiating PSMDet’s efficacy.

Table 3 demonstrates that RLK-Net-S surpasses RLK-Net-T in both depth and width, resulting in a 0.82% advantage in the mAP metric. Consequently, RLK-Net-S was ultimately chosen as the backbone network for this investigation.

Next, we determined the efficacy of the structural reparameterization ablation study. To fairly compare with the SFE block, two variants with the same number of parallel branches and without AC were designed: (A) with the same convolutional kernel size and (B) with the same equivalent convolutional kernel size. The SFE block has the highest mAP, and the mAPs of variants A and B are 0.76% and 0.94% lower, respectively, which indicates that the SFE block can effectively capture different sparse patterns, thereby ensuring the extraction of scale-sensitive features, which is crucial for high-precision detection in the end.

An ablation study of using large convolutional kernels in the middle and upper stages, Stage 2 to Stage 4, is explored. As shown in Table 4, the best mAP was obtained with K = 13 at 90.69% in the first group of experiments. K = 15 and K = 11 came next, with 90.15% and 89.57%, respectively. This indicates that the features extracted by the lower layers of the backbone network only encode local information so that they can be combined with the high-level features extracted by the final layers of the backbone network to achieve better detection.

In the second group of experiments, when all four stages use LK blocks with K = 13, the mAP is the worst, only 88.93%, which is first because Stage 1 is mainly used for extracting the low-level features such as colors, edges and textures and thus is not suitable for using a fast LK such as K = 13. In remote sensing images, the scale of objects varies greatly, and using an LK block may introduce additional background noise that drowns out relevant features such as colors, edges and textures of small objects. Secondly, the AC layer in the SFE block further makes the space of relevant features such as color, edge and texture sparse.

In addition, Table 5 also shows that replacing the large convolutional kernels in the last three stages with 3×3 kernels in the third set of experiments reduces model performance. Therefore, we use 13×13 convolutional kernels by default in the intermediate and advanced stages.

Evaluation of the efficacy of employing SK blocks to regulate network depth scaling. To explore effectiveness, we take a novel perspective and focus on the role of small convolutional kernels in controlling network depth scaling. Traditionally, expanding convolutional networks with large convolutional kernels follows a set pattern: deepening the model by simply stacking more large convolutional kernels, as in ConvNeXt-S. However, we question this and argue that adding more large convolutional kernels does not lead to a consistent performance improvement. To verify this, we designed a set of experiments, as shown in Table 6, in which we attempted to expand the number of blocks in Stage 3 from 9 to 27 based on ConvNeXt-S and explored whether these additional blocks should continue to use large convolution kernels (LK blocks).

The experimental results show that the depth expansion brings the expected performance improvement, indicating that the nine LK blocks have laid a solid foundation for performance. However, although the configuration of 27 LK blocks has a slight advantage in mAP, its inference speed has dropped significantly. In addition, removing the 3×3 convolution in the SK block significantly decreases mAP, while the throughput improvement is minimal. These findings highlight the importance of the small convolution kernels in the SK block during depth expansion. Although they may not be as effective as large convolutional kernels in expanding the ERF, small convolutional kernels provide the network with additional performance gains by increasing the spatial pattern’s abstraction level. Therefore, when building deeper convolutional networks with large convolutional kernels, a clever combination of large and small ones is the key to achieving a balance of performance and efficiency.

Next, we conducted an ablation study on the effectiveness of APN. This study incrementally incorporates various attention modules into the baseline model to assess their impact on performance. ${\mathrm{Attn}}_{L}\left(\cdot \right)$, ${\mathrm{Attn}}_{S}\left(\cdot \right)$, and ${\mathrm{Attn}}_{C}\left(\cdot \right)$ denote the scale-perception attention module, spatial-perception attention module, and task-perception module, respectively.

Table 7 illustrates that the incorporation of any module markedly enhances the mAP index, with the spatial perception attention module demonstrating the most substantial improvement, yielding a 0.78% increase in mAP. The introduction of both scale-perception and spatial-perception modules results in a further enhancement of mAP by 0.95%. The comprehensive APN module enhanced the mAP of the baseline model by 1.38%. This result illustrates the effective synergy among the modules, which collectively augment the self-modulation capability and enhance the model’s detection performance via feature alignment.

For an ablation study of regression loss functions and associated hyperparameters, this paper examines three kinds of regression loss functions based on RE: the original RE, the nonlinearly transformed f(RE), and the adjusted loss $L_{reg}\left(f\left(\mathrm{RE}\right),\tau \right)$ that includes the hyperparameter τ.

Table 8 illustrates that the performance of RE is suboptimal at 0.58% owing to its heightened sensitivity to significant errors. The implementation of basic nonlinear transformations like $\sqrt{\cdot }$ and log(·) markedly enhances performance to 88.32% and 88.86%, respectively. Furthermore, comprehensive hyperparameter optimization experiments are performed for the suggested loss function. With τ=1 and f(RE)=log(RE+1) established, the loss function attains optimal performance, yielding an mAP of 90.69%.

Last, we conducted an ablation study of normalization techniques. To mitigate the excessive increase of the regression error and obtain the normalization effect, we implemented Equation (16). Nonetheless, this supplementary normalizing procedure prompts inquiries over whether the RE genuinely enhances the results or simply adds extraneous noise. To validate our method and eliminate potential influence from normalization, we additionally normalized the ${\mathrm{smooth}}_{L_{1}}$ loss. Table 9 illustrates a substantial decline in performance following the normalizing process. This result suggests that the efficacy of RE stems not from the normalization procedure in Equation (16) but from the novel enhancement of our methodology.

### 4.5. Comparative Experiments

In the comparative trials, we selected RLK-Net-S as the foundational network.

The experimental findings in Table 10 demonstrate that our strategy achieves the highest mAP score of 90.69% on the HRSC2016 dataset. Initially, we eliminated the strategy of conventionally utilizing numerous redundant preset anchors and instead implemented a streamlined approach by deploying a singular horizontal anchor at each feature grid point. This approach markedly enhances inference speed and illustrates that reducing the number of anchor points can yield better detection results in the OD problem. The detection data presented in Figure 9 confirm the efficacy of our method. The detection results indicate no substantial center point offset or angular deviation, demonstrating that our suggested ${\mathrm{smooth}}_{\mathrm{RE}}$ loss function successfully assures the coupling of all parameters, hence achieving improved model convergence. Specifically, in the detection results numbered 3, 5, and 6, multiple scale objects are precisely identified, which can be ascribed to the proficient performance of the backbone network RLK-Net in scale-sensitive feature extraction, which is complemented by the ${\mathrm{smooth}}_{\mathrm{RE}}$ loss function and the effective feature calibration of the APN across three dimensions. In conclusion, our suggested progressive self-modulation process exhibits exceptional precision in OD and location.

The experimental findings presented in Table 11 demonstrate that our technique attains a remarkable mAP score of 89.86% on the UCAS-AOD dataset, significantly surpassing previous detectors. Figure 10 illustrates that our technique excels at recognizing intricate scenes, demonstrating significant resilience and precision. In Results 1 and 6, densely clustered little objects are precisely identified, demonstrating the efficacy and precision of our method in addressing dense objects. In Result 2, the trajectories of all vehicles on the circular lane are precisely tangent to the lane, illustrating the precision and efficacy of our approach in managing intricate traffic situations. Moreover, in Results 3, 4, and 5, our detector demonstrates resilience against leaking, even when individual automobiles are situated in building shadows or partially obscured by trees, further affirming the robustness of our methodology. Particularly in Results 6 to 9, despite encountering aircraft with unique aerodynamic configurations that blend with the background texture, our method consistently demonstrates the capability to accurately ascertain the positions and orientations of all aircraft, thereby reinforcing the robustness of our approach in managing complex objects.

Then, we used the Precision–Recall curves on the HRSC2016 and UCAS-AOD datasets (shown in Figure 11) to visualize the performance of the PSMDet model in terms of Precision rate under different Recall rates. It can be seen that PSMDet can maintain high Precision (mAP) while ensuring a high Recall rate, thus demonstrating the robustness of PSMDet in processing ORSIs with multiple scales and multiple objects.

## 5. Conclusions and Future Works

The core of the progressive self-modulation detector (PSMDet) proposed in this study lies in its unique self-modulation mechanism that runs through the backbone network, the FPN, and the regression loss function. First, RLK-Net is introduced as the backbone network in the study, and the ERF is expanded by combining the structural reparameterization technique. This design ensures the backbone can extract robust features sensitive to object scale, laying a solid foundation for achieving high-precision OD. Then, the study optimizes the multi-scale feature fusion of the FPN. By introducing a self-attention mechanism, APN enhances its ability to perceive scale changes, deeply mine spatial information, and comprehensively understand the detection task. This innovative design dramatically improves the expressive performance of the FPN, making PSMDet more efficient when dealing with complex scenes and multi-scale objects. In addition, the study also uses a GD-based BBox representation and a ${\mathrm{smooth}}_{\mathrm{RE}}$ regression loss function to achieve full coupling of the parameters and solve the regression misalignment problem in the detection of rotated objects, significantly improving the detection accuracy. Finally, using self-modulation mechanisms has enabled PSMDet to achieve an ideal balance between detection speed and accuracy. Experimental verification on the HRSC2016 and UCAS-AOD datasets has demonstrated its excellent performance.

Beyond RS, the principles and architecture of PSMDet hold significant potential for applications in intelligent transportation systems (e.g., vehicle and pedestrian detection) and industrial machine vision tasks (e.g., defect detection and quality inspection). Its robustness in handling multi-scale, arbitrary-orientation objects makes it well suited for these domains. Future research will be broadened in two dimensions: first, to thoroughly investigate the label assignment strategy for ROD to enhance detection accuracy; and second, to examine the application of anchor-free detectors to offer additional solutions and approaches for ROD.

## Figures and Tables

**Figure 1 sensors-25-01285-f001:**
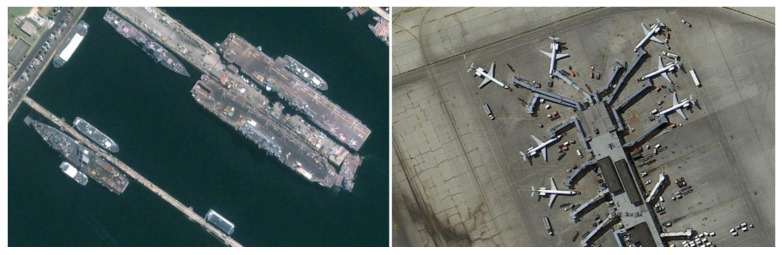
Morphology and distribution characteristics of objects with arbitrary orientation, multi-scale, and high aspect ratio.

**Figure 2 sensors-25-01285-f002:**
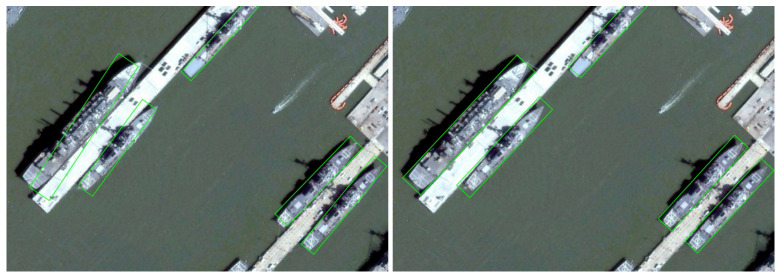
Left: Visual comparison of detection results between the ${\mathrm{smooth}}_{L_{1}}$ loss. Right: The proposed ${\mathrm{smooth}}_{\mathrm{RE}}$ loss.

**Figure 3 sensors-25-01285-f003:**
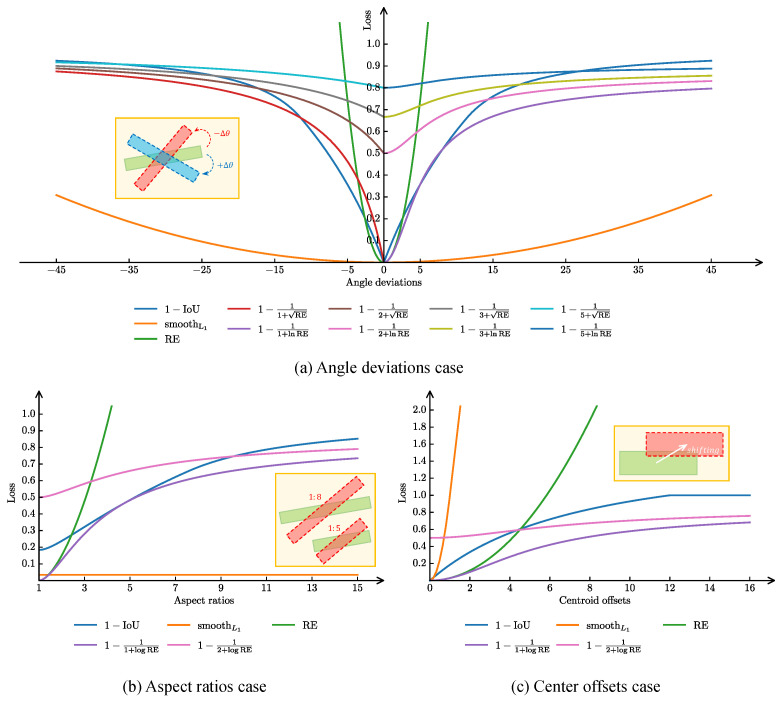
Comparison of the trends of each loss function under different parameter settings (only focusing on the ${\mathrm{smooth}}_{L_{1}}$ loss and the 1-IoU loss here [29,30]). (**a**) Relationship between angular deviation and loss function: While all loss functions possess monotonicity, only the ${\mathrm{smooth}}_{L_{1}}$ loss function is convex. (**b**) Change of loss function under different aspect ratio conditions: Under different aspect ratio settings, the change of ${\mathrm{smooth}}_{L_{1}}$ loss function stays constant, and it is difficult to perceive the change of aspect ratio. (**c**) The effect of centroid offset on the loss function: each loss function behaves consistently in terms of monotonicity, but there are differences in consistency. Even when the offset is very small, the ${\mathrm{smooth}}_{L_{1}}$ loss function tends to surge, while the 1-IoU loss is constant at 1 when the offset is so large that the predicted box does not cross the ground truth (GT) box.

**Figure 4 sensors-25-01285-f004:**
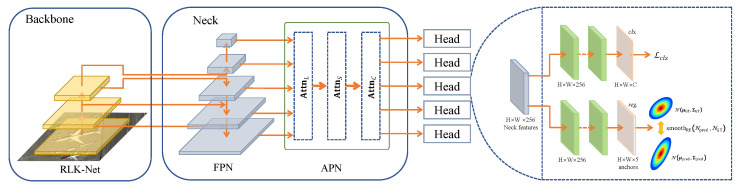
Diagram of PSMDet network architecture.

**Figure 5 sensors-25-01285-f005:**
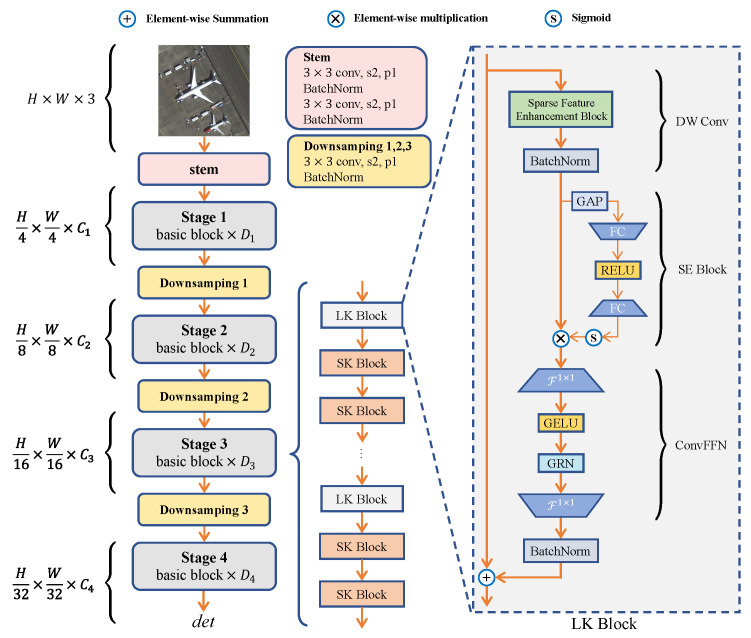
Architectural design of RLK-Net with details of the core component LK block.

**Figure 6 sensors-25-01285-f006:**
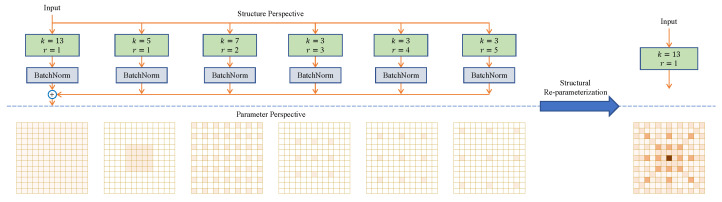
The SFE block employing a small kernel AC layer to enhance the properties of the non-atrous large-kernel convolutional layer. It shows how a parallel convolutional structure can achieve the feature extraction capability of a larger ERF.

**Figure 7 sensors-25-01285-f007:**
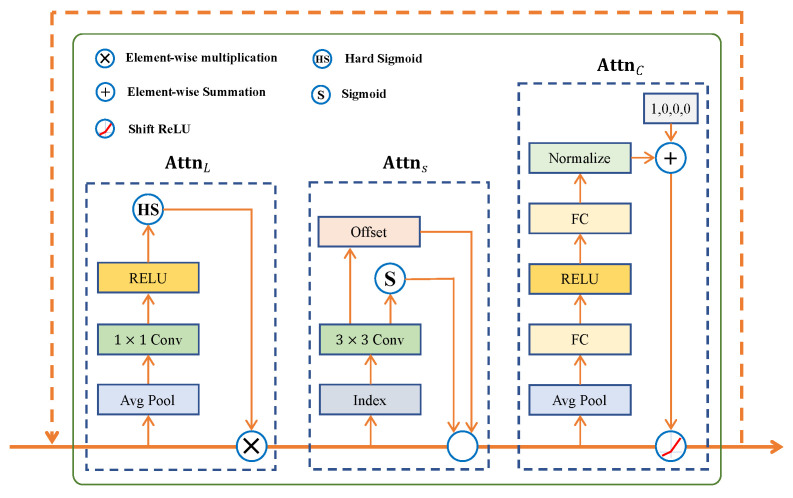
Visualization of the detailed architecture of the APN, showcasing its core design elements and operational structure.

**Figure 8 sensors-25-01285-f008:**
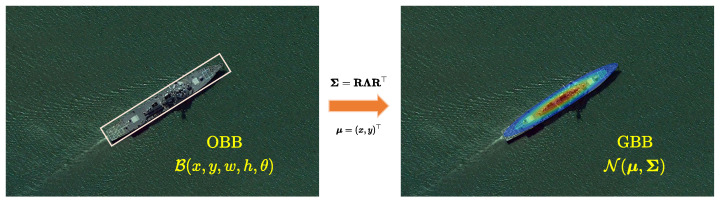
Left: Oriented bounding box B(x,y,w,h,θ). Right: Gaussian bounding box $N\left(\mu ,\Sigma \right)$.

**Figure 9 sensors-25-01285-f009:**
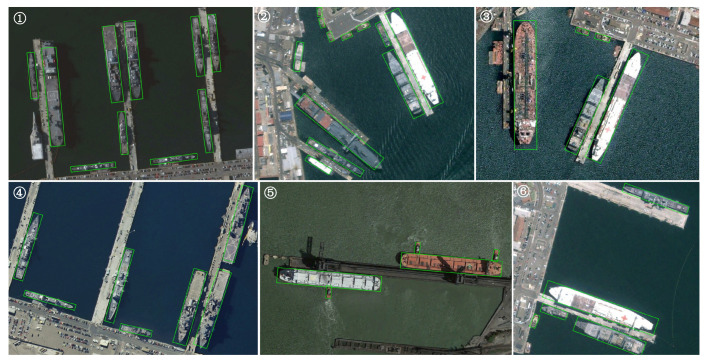
Visual representation of the conclusive test results of our methodology on the HRSC2016 dataset. For ease of description, a circled number is marked in the upper left corner of each result. Given the excellent resolution of the image graphs, it is advisable to zoom in to observe the details more clearly.

**Figure 10 sensors-25-01285-f010:**
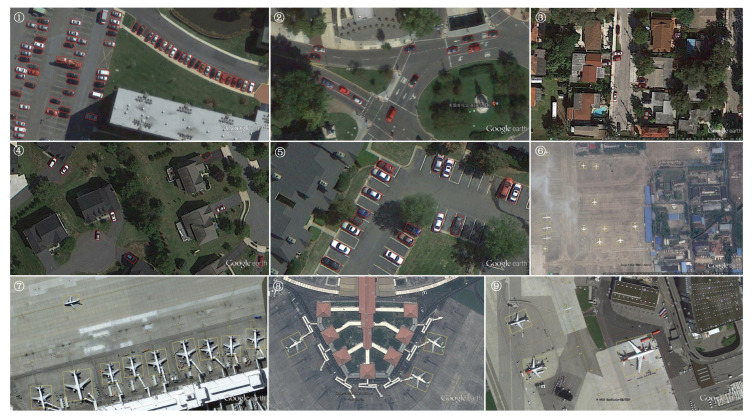
Visual representation of the ultimate test results of our methodology on the UCAS-AOD dataset across two categories. For ease of description, a circled number is marked in the upper left corner of each result. It is advisable to zoom in on the high-resolution image graphs to observe the intricacies more clearly.

**Figure 11 sensors-25-01285-f011:**
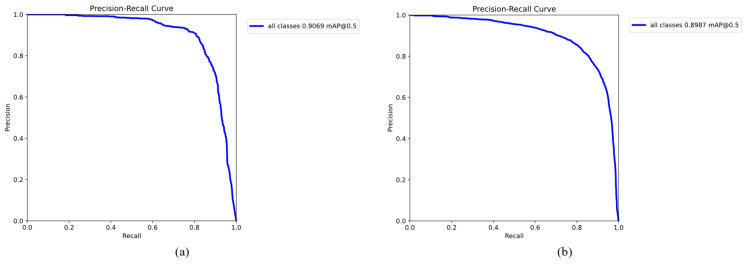
Precision–Recall curves. (**a**) HRSC2016 dataset (**b**); UCAS-AOD dataset.

**Table 1 sensors-25-01285-t001:** Architectural hyperparameters for the two RLK-Net variants (T/S) utilized in this study, specifying the number of fundamental building blocks $\left\{D_{1},D_{2},D_{3},D_{4}\right\}$ and the corresponding channel sizes $\left\{C_{1},C_{2},C_{3},C_{4}\right\}$ across the four stages.

Model	{$C_{1},\;C_{2},\;C_{3},\;C_{4}$}	{$D_{1},\;D_{2},\;D_{3},\;D_{4}$}	#P
RLK-Net-T	{80,160,320,640}	{3,3,9+9,3}	31.0 M
RLK-Net-S	{96,192,384,768}	{3,3,9+18,3}	55.6 M

**Table 2 sensors-25-01285-t002:** Results of the ablation study examining various combinations of PSMDet components on detection efficacy.

	Baseline	Different Versions of SSADet		
RLK-Net	w/o	w/	w/	w/
AAN	w/o	w/o	w/	w/
${\mathrm{smooth}}_{\mathrm{RE}}$	w/o	w/o	w/o	w/
mAP (%)	85.37	87.54	88.76	90.69

**Table 3 sensors-25-01285-t003:** Comparison of final mAP utilizing RLK-Net-S and RLK-Net-T as backbone networks.

Model	{$C_{1},\;C_{2},\;C_{3},\;C_{4}$}	{$D_{1},\;D_{2},\;D_{3},\;D_{4}$}	#P	mAP (%)
RLK-Net-T	{80,160,320,640}	{3,3,9+9,3}	31.0 M	89.87
RLK-Net-S	{96,192,384,768}	{3,3,9+18,3}	55.6 M	90.69

**Table 4 sensors-25-01285-t004:** Comparison of the final mAP for various forms of structural reparameterization applied to 13×13 convolutional layers.

Re-Param	*k*	*r*	mAP (%)
None	N/A	N/A	88.17
Same kernel size	5,7,3,3,3	1,1,1,1,1	89.75
Same equivalent kernel size	5,13,7,9,11	1,1,1,1,1	89.93
SFE	5,7,3,3,3	1,2,3,4,5	90.69

**Table 5 sensors-25-01285-t005:** Final mAP comparison using convolution kernels of different sizes in the four stages from Stage 1 to Stage 4.

Group	$D_{1}$	$D_{2}$	$D_{3}$	$D_{4}$	Params	FLOPs	mAP (%)
	3	11	11	11	32.6 M	4.86 G	89.57
1	3	13	13	13	32.9 M	4.92 G	90.69
	3	15	15	15	33.3 M	4.99 G	90.15
2	13	13	13	13	33.0 M	5.06 G	88.93
	3	3	13	13	32.8 M	4.85 G	90.16
3	3	13	3	13	32.4 M	4.81 G	90.07
	3	13	13	3	32.5 M	4.90 G	89.92

**Table 6 sensors-25-01285-t006:** Performance of different numbers of LK blocks and SK blocks in Stage 3.

$D_{3}$	LK	SK	Params	FLOPs	FPS	mAP (%)
9	9	0	32.9 M	4.92 G	15.80	89.24
27	27	0	56.7 M	9.31 G	9.71	90.71
27	14	13,3×3	55.9 M	9.15 G	10.42	90.47
27	9	18,3×3	55.6 M	9.10 G	10.72	90.69
27	9	18, w/o 3×3	55.5 M	9.08 G	10.93	89.86

**Table 7 sensors-25-01285-t007:** Results of effectiveness ablation trials for each attention module in the APN.

${\mathrm{Attn}}_{L}$	${\mathrm{Attn}}_{S}$	${\mathrm{Attn}}_{C}$	mAP (%)
w/o	w/o	w/o	87.54
w/	w/o	w/o	87.83
w/o	w/	w/o	88.32
w/o	w/o	w/	87.93
w/o	w/	w/	88.57
w/o	w/	w/	88.05
w/	w/	w/o	88.49
w/	w/	w/	88.76

**Table 8 sensors-25-01285-t008:** Results of the ablation study regarding the loss function structure and its hyperparameters on the HRSC2016 dataset.

Loss	RE	f(RE)	$L_{G}\left(f\left(\mathrm{RE}\right),\tau \right)$			
			τ=1	τ=2	τ=3	τ=5
$f\left(\mathrm{RE}\right)=\sqrt{\mathrm{RE}}$	5.80	88.32	89.93	89.15	80.35	78.45
f(RE)=log(RE)		88.86	90.69	88.76	85.91	78.67

**Table 9 sensors-25-01285-t009:** Results of ablation experiments utilizing normalizing techniques.

Loss	Norm by Equation (16)	HRSC2016	UCAS-AOD
		mAP (%)	mAP (%)
${\mathrm{smooth}}_{L_{1}}$	w/o	84.78	83.64
	w/	89.96	89.37

**Table 10 sensors-25-01285-t010:** Comparative analysis of mAP metrics on the HRSC2016 dataset.

Method	Backbone	Input_SIZE	mAP (%)
R2CNN [25]	ResNet101	800 × 800	73.07
RC1 and RC2 [53]	VGG16	-	75.7
AxisLearning [56]	ResNet101	800 × 800	78.15
Rotated RPN [12]	ResNet101	800 × 800	79.08
TOSO [57]	ResNet101	800 × 800	79.29
RRD [58]	VGG16	800 × 800	84.30
RolTransformer [13]	ResNet101	512 × 800	86.20
RSDet [28]	ResNet50	800 × 800	86.5
GlidingVertex [59]	ResNet101	512 × 800	88.20
OPLD [60]	ResNet50	1024 × 1333	88.44
BBoxAVectors [61]	ResNet101	608 × 608	88.60
DAL [62]	ResNet101	416 × 416	88.95
RIDet-Q [63]	ResNet101	800 × 800	89.10
$R^{3}$Det [14]	ResNet101	800 × 800	89.26
ACL [64]	ResNet101	800 × 800	89.46
SLA [65]	ResNet101	768 × 768	89.51
CSL [66]	ResNet50	800 × 800	89.62
RIDet-O [63]	ResNet101	800 × 800	89.63
CFC-Net [67]	ResNet101	800 × 800	89.70
GWD [22]	ResNet101	800 × 800	89.85
TIOE-Det [68]	ResNet101	800 × 800	90.16
$S^{2}$A-Net [15]	ResNet101	512 × 800	90.17
PSMDet (Ours)	RLK-Net (Ours)	800 × 800	90.69

**Table 11 sensors-25-01285-t011:** Comparison results of mAP metrics on the UCAS-AOD dataset.

Method	Backbone	Input_Size	Car	Airplane	mAP (%)
R-Yolov3 [69]	Darknet53	800 × 800	74.63	89.52	82.08
R-RetinaNet [9]	ResNet50	800 × 800	84.64	90.51	87.57
Faster RCNN [70]	ResNet50	800 × 800	86.87	89.86	88.36
RolTransformer [13]	ResNet50	800 × 800	88.02	90.02	89.02
RIDet-Q [63]	ResNet50	800 × 800	88.50	89.96	89.23
SLA [65]	ResNet50	800 × 800	88.57	90.30	89.44
CFC-Net [67]	ResNet50	800 × 800	89.29	88.69	89.49
TIOE-Det [68]	ResNet50	800 × 800	88.83	90.15	89.49
RIDet-O [63]	ResNet50	800 × 800	88.88	90.35	89.62
PSMDet (Ours)	RLK-Net (Ours)	800 × 800	88.98	90.57	89.78

## Data Availability

The HRSC2016 is available at following https://aistudio.baidu.com/aistudio/datasetdetail/31232 (accessed on 18 May 2024). The UCAS-AOD is available at following https://aistudio.baidu.com/datasetdetail/70265 (accessed on 8 May 2024).

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
