# Peer review of "PSMDet: Enhancing Detection Accuracy in Remote Sensing Images Through Self-Modulation and Gaussian-Based Regression"

_sensors, 2025, doi:10.3390/s25051285_

Round 1
Reviewer 1 Report
Comments and Suggestions for Authors
The paper presents the Progressive Self-Modulation Detector (PSMDet) framework, a new object detection method to improve detection precision in optical remote sensing images (ORSIs). The approach overcomes some challenges in rotated object detection (ROD), particularly those related to multi-scale feature representation, object orientation variations, and high aspect ratios. The framework integrates self-modulation mechanisms at three key levels to address these issues: the backbone network, feature pyramid network (FPN), and detection head.
To solve the challenges of the traditional detection paradigm in the field of RSOD, the PSMDet framework incorporates a reparameterized large kernel network (RLK-Net) for multi-scale feature extraction, an Adaptive Perception Network (APN) utilizing self-attention for feature alignment, and a Gaussian-based bounding box representation with a novel smoothRE regression loss.
The model's validation accuracy employing HRSC2016 and UCAS-AOD datasets achieved mAP scores of 90.69% and 89.86%, respectively.
The introduction and related work sections are comprehensive and contextualized well within the existing literature.
The methodology is detailed, with supporting diagrams that improve understanding of the model architecture.
However, the authors need to address some issues:
· The paper claims that PSMDet improves detection accuracy significantly, but the improvement over state-of-the-art methods is relatively small (~1-2% increase in mAP). The authors should clarify whether these small improvements translate into substantial real-world impact, referring, for example, to practical deployment scenarios.
· While the methodology is thoroughly detailed, some parts require further clarity. For example, the mathematical derivation of smoothRE loss is complex and could benefit from more intuitive explanations that could increase paper readability.
· The authors employed mAP as the primary metric. This metric provides information on but false positive/false negative trade-offs and model inference time versus accuracy trade-off. It could be interesting include precision-recall curves to provide deeper insight into performance variations.
Comments on the Quality of English Language· A thorough English language revision could enhance the readability of the paper.
Reviewer 2 Report
Comments and Suggestions for Authors
The article focuses on the development of a novel object detection system called Progressive Self-Modulating Detector (PSMDet), which addresses the challenges of traditional methods in optical remote sensing imaging, such as target complexity, scale differences and object orientation. The system uses self-modulation mechanisms, a large kernel reparameterised network to improve multi-scale feature extraction and an adaptive perceptual network for accurate feature alignment. Gaussian model based methods are also applied to solve boundary regression problems. Reviewer found that the authors have done an interesting study. There are a few comments below.
1. Line 21: Typo: the sentence should start with a capital letter.
2. Line 235: Spatial resolution of Backbone features is restored using up-sampling operation. Which upsampling method is used (bilinear, transformer, neighbourhood-based, etc.)? Does the choice of method affect accuracy?
3. Line 257: In general, traditional CNNs have a fixed receptive field, which limits their ability for global perception. However, there are architectures (e.g., dilated convolutions, attention-based mechanisms) that partially address this problem. Can RLK-Net be compared to such approaches? It is necessary to specify that.
4. Line 268: The number of channels increases in a {C, 2C, 4C, 8C} pattern. How does this relate to existing backbone architectures (e.g. ResNet, EfficientNet)? Is there any analysis of the efficiency of such channel allocation?
5. Line 273: The authors point out that BN is preferred over LN because it can be integrated into the convolutional layer, reducing the computational cost. However, LN performs better for small batches. Has a trade-off between accuracy and efficiency been considered, such as using GroupNorm?
6. Line 302: Check the reference to the source of the literature.
7. Line 320: What is the impact of moving to a non-converged AC layer on computational complexity? Does the output speed improve or deteriorate compared to traditional convolutional layers?
8. Line 356: Kernels are augmented with zeros to match a convolutional layer with a larger kernel. Can such augmentation negatively affect the quality of extracted features? For example, can it increase the blurring of object boundaries?
9. Line 520: The threshold for positive examples is set to 0.5. Have alternative values (e.g. 0.4 or 0.6) been tested? How does this affect the balance between precision and recall?
10. Line 559: The difference in mAP between K=13 and K=15 (90.69 percent and 90.15 percent) is relatively small. Has the statistical significance of this difference been tested? Is it possible that the differences are due to random factors?
11. Line 564: Is it possible that the degradation of accuracy at K=13 at all stages is due not only to the size of the kernels but also to the sparsity of the feature space?
12. Line 582: How do you interpret the decrease in mAP when the network depth is increased from 9 to 27 LK blocks? Is this a result of inefficiency in adding blocks or is it a problem in the architecture of these blocks?
13. Line 587: What exactly is it about small convolutional kernels (e.g., 3×3 or smaller) that makes them useful for regulating network depth scaling? Is there a theoretical justification for how small kernels improve spatial pattern abstraction?
Round 2
Reviewer 1 Report
Comments and Suggestions for Authors
Dear Authors,
Thank you for satisfactorily addressing all my previous comments and suggestions. Your responses have clarified the points I raised, and the improvements in the manuscript enhance its clarity and overall quality. I have no further suggestions, and I believe your paper is now suitable for publication in its present form.